# Structural characterization and evaluation of antimicrobial and cytotoxic activity of six plant phenolic acids

Monika Kalinowska[1]*, Renata Świsłocka[1], Elżbieta Wołejko[1], Agata Jabłońska-Trypuć[1], Urszula Wydro[1], Maciej Kozłowski[1], Kamila Koronkiewicz[1], Jolanta Piekut[2], Włodzimierz Lewandowski[1]

1 Department of Chemistry, Biology and Biotechnology, Institute of Environmental Engineering and Energetics, Faculty of Civil Engineering and Environmental Sciences, Bialystok University of Technology, Bialystok, Poland, 2 Department of Agri-Food Engineering and Environmental Management, Institute of Environmental Engineering and Energetics, Faculty of Civil Engineering and Environmental Sciences, Bialystok University of Technology, Bialystok, Poland

* m.kalinowska@pb.edu.pl

**Data Availability Statement:** Relevant data are within the paper and Supporting Information files. Relevant data may also be accessed via the Bialystok University of Technology repository at

## Abstract

Phenolic acids still gain significant attention due to their potential antimicrobial and cytotoxic properties. In this study, we have investigated the antimicrobial of six phenolic acids, namely chlorogenic, caffeic, *p*-coumaric, rosmarinic, gallic and tannic acids in the concentration range 0.5–500 μM, against *Escherichia coli* and *Lactobacillus rhamnosus*. The antimicrobial activity was evaluated using the 3-(4,5-dimethylthiazol-2-yl)-2,5-diphenyltetrazolium bromide colorimetric assay. Additionally, the cytotoxic effects of these phenolic acids on two cancer cell lines, the colorectal adenocarcinoma Caco-2 cell line and Dukes' type C colorectal adenocarcinoma DLD-1 cell line was examined. To further understand the molecular properties of these phenolic acids, quantum chemical calculations were performed using the Gaussian 09W program. Parameters such as ionization potential, electron affinity, electronegativity, chemical hardness, chemical softness, dipole moment, and electrophilicity index were obtained. The lipophilicity properties represented by logP parameter was also discussed. This study provides a comprehensive evaluation of the antimicrobial and cytotoxic activity of six phenolic acids, compounds deliberately selected due to their chemical structure. They are derivatives of benzoic or cinnamic acids with the increasing number of hydroxyl groups in the aromatic ring. The integration of experimental and computational methodologies provides a knowledge of the molecular characteristics of bioactive compounds and partial explanation of the relationship between the molecular structure and biological properties. This knowledge aids in guiding the development of bioactive components for use in dietary supplements, functional foods and pharmaceutical drugs.

## Introduction

There has been a growing interest in natural compounds as potential pharmaceutics or chemical agents including antioxidant or preservatives that can be applied in food industry. One of

DOI: 10.24427/2018/29/b/nz9/01997_pb (https://bazawiedzy.pb.edu.pl/info/researchdata/BUT1d8312e294574d51aee166b78f760993/).

**Funding:** This research was funded by The Polish Ministry of Education and Science, project number WZ/WB-IIS/3/2020 and National Science Center no. 2018/29/B/NZ9/01997.

**Competing interests:** The authors have declared that no competing interests exist.

the known natural bioactive substances are phenolic compounds, which include flavonoids, phenolic acids, or tannins [1, 2]. Phenolic compounds are secondary metabolites that are involved primarily in protecting plants against UV radiation, viruses, bacteria and insects. In the past decade, scientific publications taking a closer look at phenolic compounds have increasingly highlighted their antibacterial and cytotoxic properties and the potential for their use. Hence, there is a great deal of co-occurrence of these keywords in many global studies (S1 Fig) [3]. Phenolic acids are among the most widely distributed plant non-flavonoid phenolic compounds present in the free, conjugated-soluble and insoluble-bound forms.

Chlorogenic acid (5-CQA) (S2 Fig) [4], an ester of caffeic acid and (−)-quinic acid, one of the most abundant phenolic acids, is widely distributed in fruits and vegetables like: apples, pears, carrots, or potatoes. Additionally, it can be found in most consumed beverages in the world: wine, tea and coffee [5–7]. Notably, chlorogenic acid demonstrates antimicrobial activity against bacteria (S1 Table) like Gram-negative *Escherichia coli* (MIC 7.06 mg/mL) [8], Gram-positive *Alicyclobacillus acidoterrestris* (MIC 2 µg/mL) [9] and fungi *Candida albicans* with MIC (80 µg/mL) [10]. 5-CQA exhibits antimicrobial activity by disrupting membrane, leading to permeability changes and release of cytoplasmic macromolecules, resulting in cell death [11]. 5-CQA main cytotoxic effects on cancer cells are triggering apoptosis, suppressing cell proliferation, or inhibiting the cell cycle. Studies show (S2 Table) lack of cytotoxicity of 5-CQA towards HaCaT cell line in concentration range of 0.15–1000 nM [5], HL-60 and Jurkat cells in concentrations of 1–100 µM [12].

Caffeic acid (CA) (S2 Fig), is a well-studied phenolic acid present in numerous plant species. It can be found in herbs like basil, rosemary, sage, oregano and even in vegetable like carrots, potatoes, and also in propolis [13]. Its antimicrobial abilities studied by Arciszewska et al (S1 Table) showed inhibitory effect against *E. coli* (MIC 13 mg/mL), *Bacillus subtilis* (MIC 18 mg/mL) and antifungal inhibiting growth of *Candida albicans* (MIC 29 mg/mL) [14]. Another research showed that CA completely inhibits growth of *Staphylococcus aureus*, *Pseudomonas aeruginosa*, *E. coli*, *Candida albicans*, *Aspergillus brasiliensis* at concentration of 5 mM [15]. CA possesses nucleophilic properties and can donate an electron pair to plasma membrane proteins and lipids, potentially leading to membrane dysfunction. CA may physically damage the cytoplasmic membrane, resulting in the leakage of intracellular components and affecting various cellular processes [16]. CA has been investigated in various cancer cell lines (S2 Table), including MDAMB-231, SK-BR-3, HeLa, HT-29, Neuro 2A, PC-3, and Vero cells. In these cell lines, it exhibited $IC_{50}$ values ranging from 104 µg/mL to 261 µg/mL, indicating its potential cytotoxic effects against various type of human cancer cells but also animal ones [17]. CA's anticancer effects are attributed to its both antioxidant and pro-oxidant capacities, its chemical structure that possesses phenolic hydroxyl groups and a double bond in the carbon chain. These factors enable the elimination of free radicals and prevention of reactive oxygen species production as well its ability to stimulate the expression of P53 and P21 genes while inhibiting CDK2 gene expression, leads to cell cycle arrest in the G0/G1 phase and the inhibition of MMP-2 and MMP-9 expression, which are involved in cancer invasion and metastasis [18, 19].

*p*-Coumaric acid (*p*-CA) (S2 Fig) is commonly found in cherries, walnuts, onions, pear, grapes and wine either as an esterified form or as a free acid. Experimental and clinical studies have demonstrated that *p*-CA possesses a range of beneficial properties, including antioxidant and anti-inflammatory activities, making it highly valuable for industries such as food, health, cosmetics and pharmaceuticals [20, 21]. Antimicrobial properties of *p*-CA have attracted the attention of many researchers (S1 Table). For example, Bag et al. have examined *p*-CA growth inhibitory capabilities on *Bacillus cereus and Salmonella typhimurium* with results of respectively MIC equal to 0.041 mg/mL and 0.104 mg/mL [22]. In studies conducted by Abdel-

Wareth et al. *p*-CA showed no antimicrobial action against *Pseudomonas aeruginosa*, *Escherichia coli*, *Staphylococcus aureus*, *Aspergillus niger* or *Candida albicans* but exhibited ability to inhibit growth of *Bacillus cereus* (IZ = 9 mm) [23]. *p*-CA antibacterial mechanism was investigated by Lou et al. The results showed that *p*-CA caused an efflux of potassium ions from cells, indicating an increase in membrane permeability. It was revealed that *p*-CA caused membrane damage, leading to the release of cellular contents. Furthermore, their studies showed that *p*-CA can bind to DNA, potentially affecting its structure and function leading to death of a cell [24]. *p*-CA exhibits significant cytotoxic abilities against various cancer cell lines (S2 Table). *p*-CA demonstrated its great efficacy in inhibiting the growth of HeLa cells, with an $IC_{50}$ value of 1 μg/mL. HT-29 cells exhibited an $IC_{50}$ value of 25 μg/mL suggesting a substantial cytotoxic effect. Karakurt et al. evaluated cytotoxic effects of *p*-CA using the Alamar Blue assay. The study determined that *p*-CA exhibited cytotoxic abilities against DLD-1 cells with an $IC_{50}$ value of 254.3 μg/mL Caco-2 cells with $IC_{50}$ of 714.6 μg/mL [25].

Rosmarinic acid (RA) (S2 Fig) is predominantly found in herbs such as rosemary, sage, basil, oregano, marjoram, and lemon balm [26]. Some of its notable properties include neuroprotective effects, antioxidant activity, anti-inflammatory properties, antimicrobial effects, anti-angiogenic properties, heptatoprotective and anticancer potential [27–29]. RA showed notable antimicrobial activity against various microorganisms (S1 Table). RA induces membrane depolymerization upon contact. Furthermore, RA inhibits the activity of $Na^+/K^+$-ATPase in bacterial cells [19]. In terms of antifungal mechanisms, RA may hinder mitochondrial activity, compromises membrane integrity [19]. RA demonstrated an $IC_{50}$ value of 2.9 μmol/L for the MOLM-13 cell line, the MOLM-14 cell line showed an $IC_{50}$ value of 7.1 μmol/L [30]. MDAMB-231, SK-BR-3, and HeLa cells also showed notable susceptibility to RA, with $IC_{50}$ values of 121 μg/mL, 147 μg/mL, and 139 μg/mL, respectively. Similarly, the $IC_{50}$ values for RA against HT-29 and Vero cells were 150 and 388 μg/ml, respectively [17]. Numerous mechanisms have been suggested to explain the anticancer effects of RA. RA inhibits colony formation, cell viability, and spheroid creation by targeting HDAC2 [31]. This leads to cell cycle arrest and apoptosis. RA also demonstrates dose-dependent inhibition of certain cells while being less toxic to normal lymphocytes [32].

Gallic acid (S2 Fig) (GA) is the most popular of trihydroxybenzoic acids that is found in the leaves of pomegranate bark, gall nuts, Indian gooseberry, witch hazel, sumac, tea leaves, oak bark, and many other plants, both in free form and as part of tannin molecules. It occurs in abundance in processed beverages such as red wine or green tea [33]. GA exhibits significant antimicrobial abilities against a wide range of microorganisms, including bacteria and fungi (S2 Table). In studies conducted by Kalinowska et al it displayed a MIC of 3 mg/mL against *E. coli*, *B. subtilis*, *Salmonella enteritidis*, and *C. albicans*, however *S. aureus* was more resistant to GA antibacterial abilities with MIC 6 mg/mL [34]. Furthermore, it generates an IZ of 11 mm against *Pseudomonas fluorescens*, 13.2 mm against *P30-4*, 15 mm against *Pseudomonas fragi*, and 14.8 against *Pseudomonas putida* [35]. In case of fungal infections, GA demonstrated its effectiveness by inhibiting the growth of *Streptococcus mutans* and *Pseudomonas aeruginosa*, with MIC values of 62.5 μg/mL and 100 μg/mL, respectively [36]. One of GA antimicrobial mechanisms is the inhibition of efflux pumps, which contribute to antimicrobial resistance in bacteria [37]. GA might cause impairing in the permeability of cellular membranes and also altering its charge or hydrophobicity. This might lead to DNA degradation, impairment of gene expression and cell signaling, disruption of energy metabolism, and ATP depletion [38, 39]. GA has been evaluated for its cytotoxicity in various cell lines using different assays (S2 Table). In the research conducted by Pham et al. on A2780, H460, A431, MCF-7, and MCF10A cell lines using the MTT assay, the concentration of 50 μM GA did not exhibit any cytotoxic properties after 72 h of incubation [40]. However, studies by Rezaei-Seresht et al. on

breast cancer MCF-7 line showed that GA possess cytotoxic activity against this specific cell line with IC$_{50}$ value of 18 μg/mL [18]. GA has been found to increase the expression of p53 and disrupts mitochondrial membranes, which may lead to cytochrome C release, caspase-3 activation, and subsequent apoptotic cell death. Additionally, GA can mediate in DNA damage-related regulations and overexpression of the P21 gene. The P21 can promote cellular differentiation and inhibition of cell proliferation [18].

Tannic acid (TA) (S2 Fig) is a naturally occurring plant polyphenol that can be found in various plants, notably in certain tree species such as sumac, oak, and myrobalan, particularly their barks, are rich sources of it. Additionally, tannic acid can be found in galls formed on plants [41]. It has a rich history of medicinal use, including the treatment of diarrhea, burns, and rectal disorders [42]. Additionally, it has been associated with various health benefits, including modulating immune responses, reducing allergen levels, and accelerating blood clotting [42–44]. Tannic acid exhibits antibacterial abilities against various bacterial strains. The minimum inhibitory concentrations [MICs] of tannic acid in studies conducted by Sahiner et al. were as follows: 60 μg/mL for *P. aeruginosa*, 60 μg/mL for *B. subtilis*, 40 μg/mL for *S. aureus* and 70 μg/mL for *C. albicans*. However, in the same study TA expressed no antimicrobial activity against *E. coli* [44]. On the other hand, tannic acid demonstrated inhibitory effects against *E. coli* with an average diameter of 13.33 mm and against *S. aureus* with a diameter of 10.16 mm [45]. Moreover, TA exhibits a moderately inhibitory effect against *S. typhimurium* 400 μg/mL (MQIC value, minimal quorum inhibitory concentration). In case of methicillin-resistant *Staphylococcus aureus*, tannic acid showed a MIC of 64 μg/mL [46]. TA penetrates bacterial cell walls, disturbing bacterial metabolism, and potentially causing DNA leakage. Moreover, TA limits bacterial growth by hindering sugar and amino acids absorption [47]. TA demonstrates cytotoxic abilities against various cancer cell lines (S2 Table). TA induces cytotoxicity by caspase-mediated apoptosis, suppresses fatty acid synthase and inhibits epidermal growth factor receptor pathway. It reduces viability, metastatic potential and stemness of cancer cells [48–50].

In this study, we will evaluate the antimicrobial activity of these six natural phenolic acids against a bacterial strains living in the digestive system, i.e. *Escherichia coli* and *Lactobacillus rhamnosus* based on 3-(4,5-dimethylthiazol-2-yl)-2,5-diphenyltetrazolium bromide colorimetric assay. Studies of antimicrobial activity aim to determine the suitability of the tested compounds as potential preservatives with antimicrobial properties against selected food pathogens (*E.coli*). Moreover, the studies on the effect of the analysed compounds on *L. rhamnosus* were conducted, which are part of the human intestinal microflora to exclude any negative effects on beneficial probiotic bacteria. We will also investigate those phenolic acids cytotoxic effects of on two colorectal adenocarcinoma cell lines: Caco-2 and DLD-1 cell lines in order to analyze potential anticancer activity of studied compounds. Additionally, some structural parameters of the phenolic acids such as lipophilicity coefficient (logP) and reactivity of molecules on the basis of parameters calculated in Gaussian 09W program such aslike ionization potential, electron affinity, electronegativity, chemical hardness, chemical softness, dipole moment and electrophilicity index. The six natural phenolic compounds were chosen on purpose. They have proven biological properties such as antioxidant, antimicrobial or chemopreventive evaluated in different model systems of bacteria or cell lines (S1 and S2 Tables). The chosen ligands are derivatives of two important aromatic acids, i.e. benzoic and cinnamic acids which differ in the presence of vinyl moiety between the aromatic ring and carboxylic group. Moreover they differ in the number of hydroxyl substituents in the ring (i.e. *p*-coumaric, caffeic and gallic acid), some of them are caffeic acid esters (i.e. rosmarinic and chlorogenic acids) or gallic acid ester (i.e. tannic acid). All this structural difference influence on the solubility, lipophilicity and reactivity of molecules. Therefore the following questions may arise:

(1) does the number of -OH substituents in the aromatic ring is the main factor that affect the antimicrobial and cytotoxic activity of compounds (toward selected bacteria and cell lines), (2) does the conjugation of caffeic and gallic acid with other molecules influence on the physico-chemical and biological properties of molecules, (c) do the selected experimental (logP) and theoretical (e.g. energy of HOMO or LUMO orbitals) correlates with the biological activity of molecules?

## Materials and methods

### Reagents

Dulbecco's modified Eagle's medium (DMEM) with 4.5 mg/mL (25 mM) of glucose with Glutamax, penicillin, streptomycin, trypsin–EDTA, FBS (Fetal Bovine Serum) Gold, and PBS (Phosphate Buffer Saline) (without Ca and Mg) were provided by Gibco (San Diego, CA, USA). MTT reagent was purchased from Sigma-Aldrich.

### Quantum-chemical calculations

The density functional theory (DFT) calculations were performed in the gas phase with approximation of the isolated molecules using the Gaussian 09 [51] program running on PC computer and Gauss View [52] molecular visualization program. To calculate the geometrical structure of the compounds the B3LYP method with 6–311++G(d,p) basis set was used. The exception was represented by a molecule of tannic acid, where the HF631/3-21G method was employed. The optimized structures were possessed the minimum in the energy with no imaginary infrared wavenumbers.

### Bacterial strains

*Escherichia coli* (ATCC 25922) and *Lactobacillus rhamnosus* (ATCC 53103) were obtained from the American Type Culture Collection (Manassas, VA, USA). *E. coli* (Gram-negative bacteria) and *L. rhamnosus* (Gram-positive bacteria) were grown overnight in Mueller Hinton II Broth at 37˚C. The next day, the overnight cultures were diluted in fresh MH II Broth to obtain $10^8$ CFU/mL (CFU–colony forming units). For the antimicrobial activity was used the inoculum where the suspension of *E. coli* and *L. rhamnosus* cells was $10^6$ CFU/mL.

**Determining antimicrobial activity assay.**    Antimicrobial activity of tested *p*-coumaric acid (*p*-CA), caffeic acid (CA), gallic acid (GA), chlorogenic acid (5-CQA), rosmarinic acid (RA) and tannic acid (TA) against of *E. coli* and *L. rhamnosus* were calculated based on 3-(4,5-dimethylthiazol-2-yl)-2,5-diphenyltetrazolium bromide (MTT) colorimetric assay. The MTT assay was conducted according to Jablonska-Trypuc et al (2019) [53]. The final concentrations of tested compounds in each well were: 500 μM, 250 μM, 125 μM, 62.5 μM, 31.25 μM, 15.63 μM, 7.81 μM, 3.91 μM and 0.98 μM. The selected concentrations correspond to the levels of detected tested compounds in the human body as a result of the consumption of products rich in phenolic compounds.

**Cytotoxicity assay.**    The cytotoxicity assay was performed according to the method of Carmichael et al using MTT reagent [54]. Activity of 5-CQA, CA, *p*-CA, GA, TA and RA in the concentration range from 0.5μM to 500μM, was studied in colorectal adenocarcinoma Caco-2 cell line and Dukes' type C colorectal adenocarcinoma DLD-1 cell line obtained from ATCC. 5-CQA, CA, *p*-CA, RA, GA and TA were stored at temperature 4˚C and the stock solution was prepared by dissolving it in TrisHCl buffer prior analysis. GloMax®-Multi Microplate Multimode Reader was used in order to measure the absorbance (570 nm). The viability of DLD-1 cells and Caco-2 cells was calculated as a percentage of control non—treated cells [54].

## Statistical analysis

All data are given as me an values ±SD (standard deviation). Differences between treatments and untreated control human cells were analyzed by one-way ANOVA, followed by Dunnett's procedure for multiple comparisons. Significant effects are represented by $p \leq 0.05$ (*), $p \leq 0.01$ (**), $p \leq 0.001$ (***).

# Results and discussion

## The theoretical electronic parameters and lipophilicity

The calculated dipole moment, the energy of the HOMO and LUMO orbitals and the electronic parameters (such as ionization potential (I), electron affinity (A), electronegativity (χ), chemical hardness (η), softness (S), chemical potential (μ) and electrophilicity (ω) indexes calculated on the basis of the HOMO and LUMO orbital's energy) [55] were obtained for the studied compounds (Fig 1). The parameters are used to predicting the electronic charge transfer within the molecule, reactivity, and stability of the compounds (S3 Table and Table 1) [55]. Because for tannic acid the electronic parameters were obtained at different theoretical level than for the other ligands, they can't be directly compared within the whole series of the compounds.

The HOMO-LUMO gap values increase in the following order: RA < 5-CQA < CA < p-CA < GA, indicating the decrease in the bioreactivity of studied molecules in above series (Table 1). As a result, it can be inferred that among the studied compounds, rosmarinic acid is the most chemically active, while gallic acid is the least active. The ionization potential decreases and the value of the energy of HOMO orbital increases in same the series: p-CA → GA → 5-CQA → CA → RA what means the rise in the electron donating properties in the above series of ligands in the redox reactions.

The presented MEP surface (Fig 2) describes the overall molecular charge distribution and allows to predict the sites for electrophilic and nucleophilic attack. The red region in the MEP plot shows an electron-rich site with electrophilic reactivity that covers the oxygen atoms in carbonyl group and phenolic OH groups. Whereas the blue color indicates an electron-deficient region which is the site of the nucleophilic attack and it is mainly located in the region of the hydrogen atom of the phenolic OH groups.

## Lipophilicity

The logarithmic coefficient of substance distribution (log P) in the water-n-octanol system is a crucial parameter used to assess the hydrophilicity/lipophilicity of chemicals. Factors like bioaccumulation and toxicity are largely correlated with logP. Regarding most of the analyzed phenolic acids, the collected experimental data and calculated log P values are consistent with each other (Table 2). Based on the experimental logP values, the compounds can be arranged in order of increasing lipophilicity as follows: 5-CQA< GA< CA< RA< p-CA< TA. GA and 5-CQA logP experimental values of 0.70 and 0.30 respectively indicates their low lipophilicity (Table 2). LogP experimental values for RA, CA and *p*-CA are in the range of 1–2 this means that these compounds possess moderately lipophilic properties. The experimental logP value of 4.84 for TA indicates it's high lipophilic properties. pKa is a measure of the acidity or basicity of a compound. In the context of lipophilicity, pKa can provide information about the ionization state of a compound at different pH values. Compounds with lower pKa values are more likely to be ionized at physiological pH (around 7.4), while compounds with higher pKa values are more likely to be in their neutral, non-ionized form. Therefore, based on experimental data and calculated values, it can be deduced that the molecular and monodeprotonated

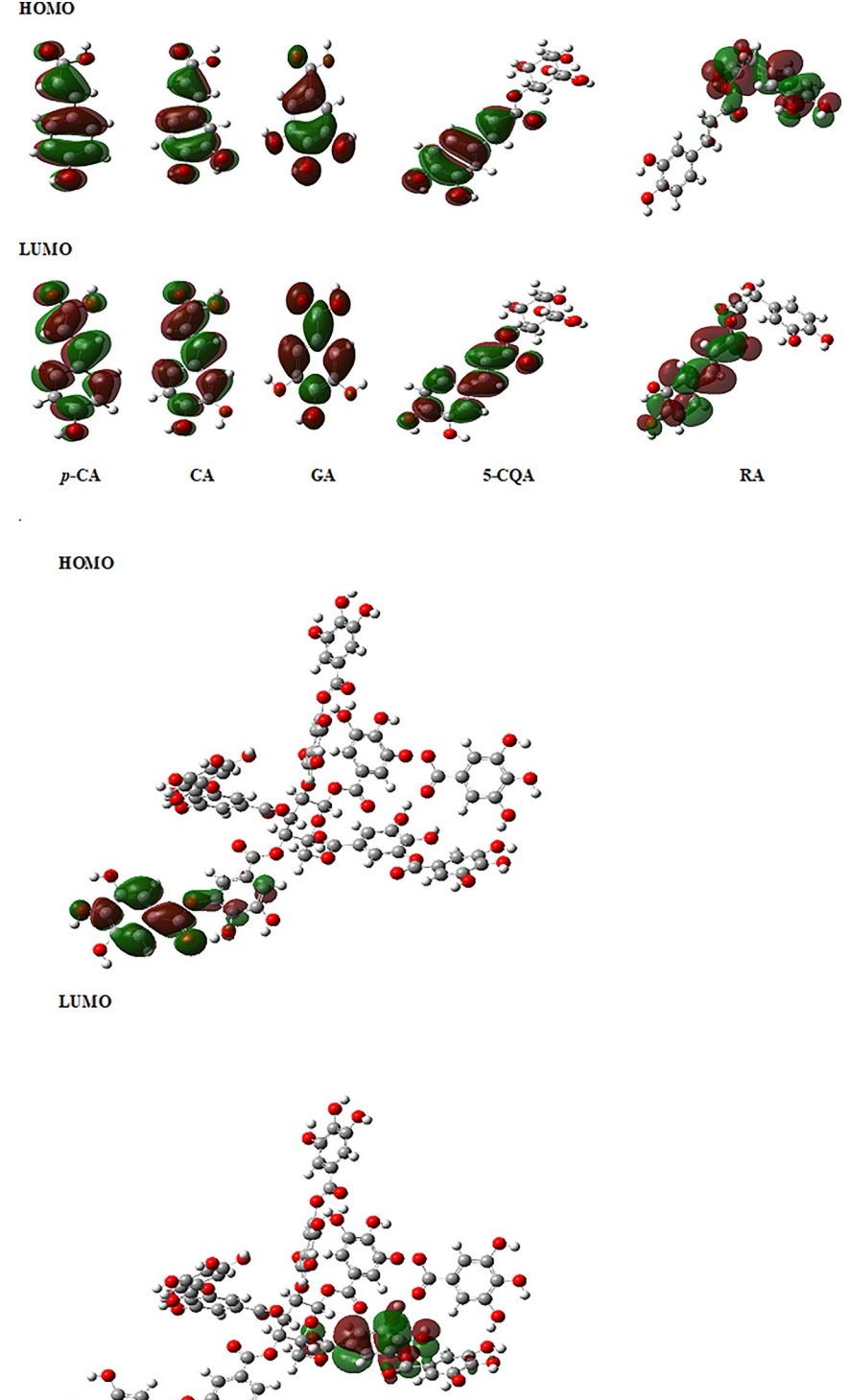

**Fig 1. HOMO and LUMO orbitals for *p*-coumaric acid (*p*-CA), caffeic acid (CA), gallic acid (GA), chlorogenic acid (5-CQA), rosmarinic acid (RA) and tannic acid (TA).**

**Table 1. Energy of HOMO/LUMO orbitals and other reactivity descriptors.**

| Descriptors | p-CA | CA | RA | 5-CQA | GA | TA |
|---|---|---|---|---|---|---|
| $E_{HOMO}$ (eV) | -6.431 | -6.252 | -5.796 | -6.281 | -6.344 | -8.554 |
| $E_{LUMO}$ (eV) | -2.148 | -2.114 | -2.251 | -2.198 | -1.613 | -1.805 |
| Energy gap (eV) | 4.283 | 4.138 | 3.545 | 4.083 | 4.731 | 6.749 |
| Ionization potential, $I = -E_{HOMO}$ | 6.431 | 6.252 | 5.796 | 6.281 | 6.344 | 8.554 |
| Electron affinity, $A = -E_{LUMO}$ | 2.148 | 2.114 | 2.251 | 2.198 | 1.613 | 1.805 |
| Electronegativity, $\chi = I + A/2$ | 4.289 | 4.183 | 4.023 | 4.240 | 3.978 | 5.179 |
| Chemical potential, $\mu = -I + A/2$ | -4.289 | -4.183 | -4.023 | -4.240 | -3.978 | -5.179 |
| Chemical hardness, $\eta = I - A/2$ | 2.141 | 2.069 | 1.773 | 2.042 | 2.366 | 3.374 |
| Chemical softness, $S = 1/2\eta$ | 0.234 | 0.242 | 0.282 | 0.245 | 0.211 | 0.148 |
| Electrophilicity index, $\omega = \mu^2/2\eta$ | 4.296 | 4.228 | 4.566 | 4.402 | 3.345 | 3.975 |

forms of GA, RA, 5-CQA, p-CA and CF are predominant within the range of physiological pH. Among those phenolic compounds 5-CQA and RA possessed lowest pKa experimental value (3.42 and 3.57 respectively), this is due to high number of the hydroxyl groups they possess. The presence of functional groups that can act as hydrogen bond donors or acceptors enhances the hydrophilicity of a compound. Compounds containing hydroxyl and carboxylic groups have the ability to form hydrogen bonds with water molecules in an aqueous environment, influencing their solubility in water. However tannic acid possesses multiple hydroxyl groups and exhibits a pKa value of 8.5. As a result, tannic acid is significantly less acidic and highly lipophilic compared to the other studied phenolic compounds. This heightened hydrophobicity and low acidity can primarily be attributed to its strong propensity to form hydrogen bonds with itself.

## Antimicrobial activity

Fig 3 shows the influence of *p*-CA, CA, GA, 5-CQA, RA and TA on the concentration range 0.98–500 μM on tested *E. coli* and *L. rhamnosus* strains. The obtained results are in agreement with S1 Table. In the case of *E. coli*, it was observed that CA, *p*-CA and 5-CQA caused an insignificant increase of the bacteria growth with rising concentration of tested acids and the highest value did not exceed 16% (Fig 3). For GA the average relative viability of *E. coli* was approximately of 1 to 10% compared to untreated cells. After the application of TA at the concentrations from 62.5 to 500 μM one can observe a relative decrease in the cell viability approximately of 1 to 10% compared to the control cells. In the studied range of concentration the selected phenolic acid showed no effect on *E. coli* [44].

On the other hand, the studied phenolic compounds had differentiated effects on *L. rhamnosus* viability. In the studied range of concentration 0.98–500 μM GA, 5-CQA, CA, RA significantly stimulated the tested bacteria strain (only in case of RA at the highest concentration 500 μM no stimulation was observed). Especially for CA, 5-CQA and RA a significant increase of *L. rhamnosus* viability with increasing concentration of the tested acids was observed and the highest value for the bacteria did not exceed 180%. These phenolic compounds possessed the lowest logP among the tested compounds which increase in the order GA→5-CQA→CA→RA (0.70–1.60). The values of the energy of the HOMO orbitals (EHOMO) calculated for these compounds increase in the same order as well. In turn, after the application of TA on L. rhamnosus at the lowest concentrations from 0.98 to 15.63 μM, an increase in relative cell viability of 40 to 60% was observed, while at TA concentrations from 31.25 to 500 μM there was a decrease in the relative cell viability of 20 to 80% compared to the control untreated cells. For p-CA the inhibitory effect on L. rhamnosus viability was observed only in the case of

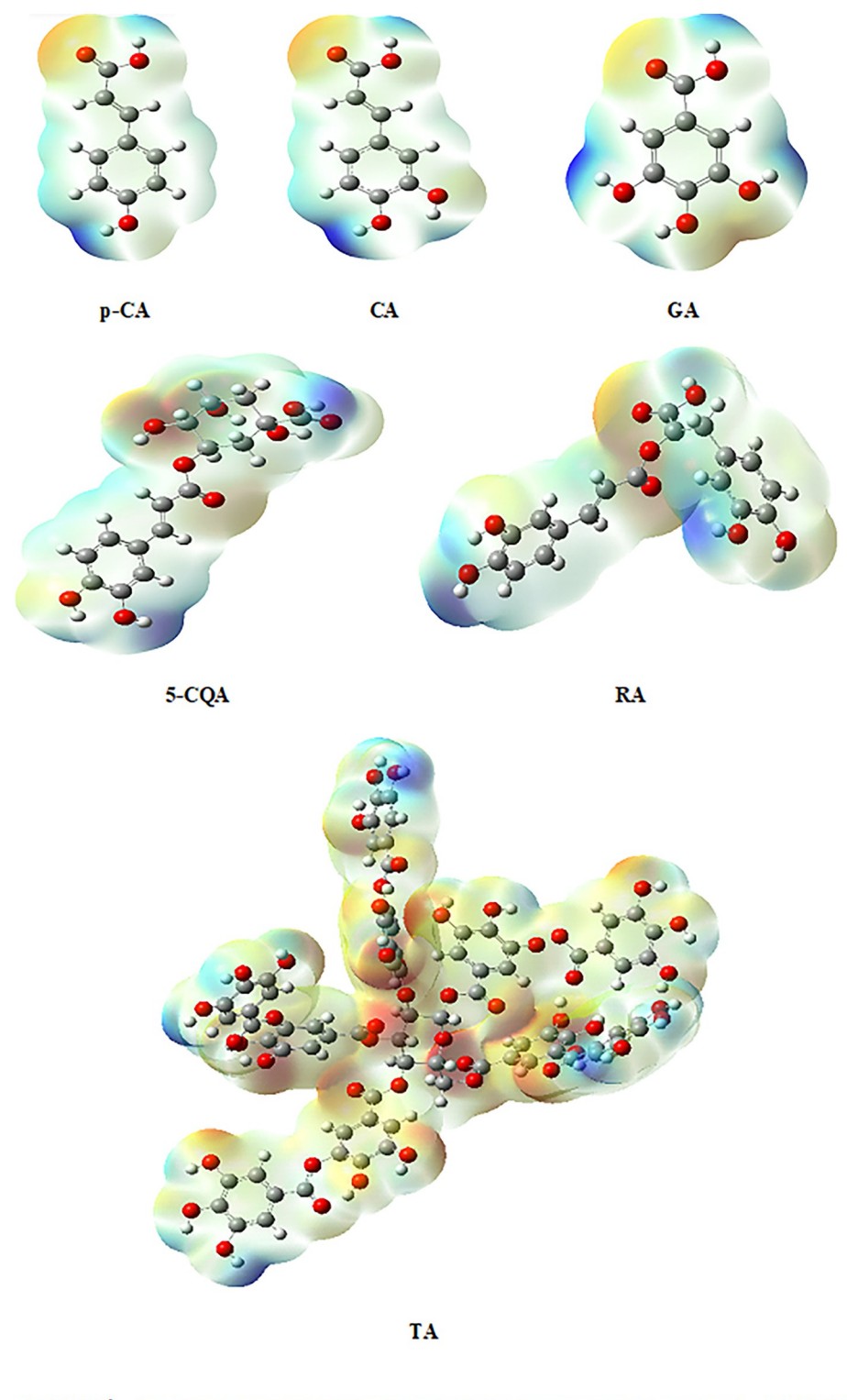

**Fig 2. Calculated electrostatic potential surfaces on the molecules for *p*-coumaric acid (*p*-CA), caffeic acid (CA), gallic acid (GA), chlorogenic acid (5-CQA), rosmarinic acid (RA) and tannic acid (TA).**

**Table 2. Theoretical (calculated in ACD/labs or percepta: LogP classic, logP galas, pKa classic) and experimental logP and pKa values.**

| Compound | ACD/LogP Classic | ACD/ LogP Galas | Conssensus P | LogP experimental | Source | ACD/pK$_a$ Classic | pKa experimental | Source |
|---|---|---|---|---|---|---|---|---|
| Gallic acid | 0.91 ± 0.33 | 0.61 | 0.69 | 0.70 | [56] | 4.3± 0.1 | 4.21 | [57] |
| 5-CQA | -0.36 ± 0.43 | -0.30 | -0.33 | 0.30 | [58] | 3.9± 0.5 | 3.42 | [58] |
| Caffeic acid | 1.42 ± 0.36 | 1.32 | 1.35 | 1.12 | [58] | 4.6± 0.1 | 4.49 | [59] |
| p-coumaric acid | 1.88 ± 0.34 | 1.63 | 1.69 | 1.79 | [60] | 4.5±0.4 | 9.50 | [61] |
| Rosmarinic acid | 1.70 ± 0.41 | 1.54 | 1.60 | 1.60 | [62] | 2.8± 0.1 | 3.57 | [63] |
| Tannic acid | 13.33 ± 0.91 | 3.93 | 9.53 | 4.84 | [64] | 6.0 ± 0.5 | 8.5 | [65] |

the concentration 500 μM. The observation correlates with the value of logP and EHOMO which are respectively the highest and which is the lowest for these two phenolic acids (comparing the rest of studied phenolic acids), i.e. for TA: logP = 4.86, -EHOMO = -8,554 eV and p-CA: logP = 1.69; EHOMO = -6.431 eV. The obtained results suggest that the phenolic compounds in the studied concentration range did not significantly influence the viability of E. coli, which indicates that these compounds do not stimulate the growth of a potentially pathogenic bacterium naturally present in the human microbiome. On the other hand GA, CA, 5-CQA, p-CA, RA possess prebiotic properties toward *L. rhamnosus* in the concentration range 0.98–250 μM. For TA this stimulatory effect was observed in the concentration range 0.98–15.63 μM.

## Cytotoxic activity

As it was shown in Figs 4 and 5, studied compounds did not exert significant cytotoxic effect on Caco-2 cells. 5-CQA caused increases in relative Caco-2 cell viability even in 200μM concentration. After 48h treatment decreases in studied parameter were observed in lower concentrations (from 0.5μM to 10μM). In case of *p*-CA no significant decreases in cell viability as compared to control untreated cells were noticed. Relatively more significant effect was observed in case of *p*-CA after 48h treatment, when decreases below control level were observed in all analyzed concentrations.

It should be underline that studied compounds were more effective against DLD-1 cells that towards Caco-2 cells (Figs 4 and 5). Statistically significant decreases in relative cell viability were observed under the influence of *p*-CA already in 0.5μM concentration. After 48h incubation 26% decline in cells viability as compared to control untreated cells was noticed. DLD-1 cells treated with RA showed decrease in cell viability already in 5μM concentration.

As shown in Figs 4 and 5, in Caco-2 cell line cytotoxic effect was observed rather after 48h incubation with both studied compounds. In DLD-1 cell line incubation with especially TA in all analyzed concentrations leads to statistically significant decreases in cell viability. However, 5μM of GA in Caco-2 cell line caused an increase in relative cell viability. Similar effect was observed in case of TA treatment after 48h incubation time in the lowest tested concentration. TA in 200 μM concentration caused almost 50% decline in Caco-2 cell viability after 48h incubation. More significant inhibitory effect of studied compounds was observed in DLD-1 cell line than in Caco-2 cell line. Decrease by about 50% was noticed in DLD-1 cells in the concentration of 200μM for both compounds. It should be also mentioned that in case of DLD-1 cell line we did not observe any increases as compared to control untreated cells. From the group of studied compounds TA, GA and RA were especially active inDLD-1 cell line. Obtained results are in accordance with literature data indicating anticancer properties of phenolic compounds [12]. Differences between the individual tested compounds may result from their chemical structure and thus their ability to penetrate cell membranes. The research results

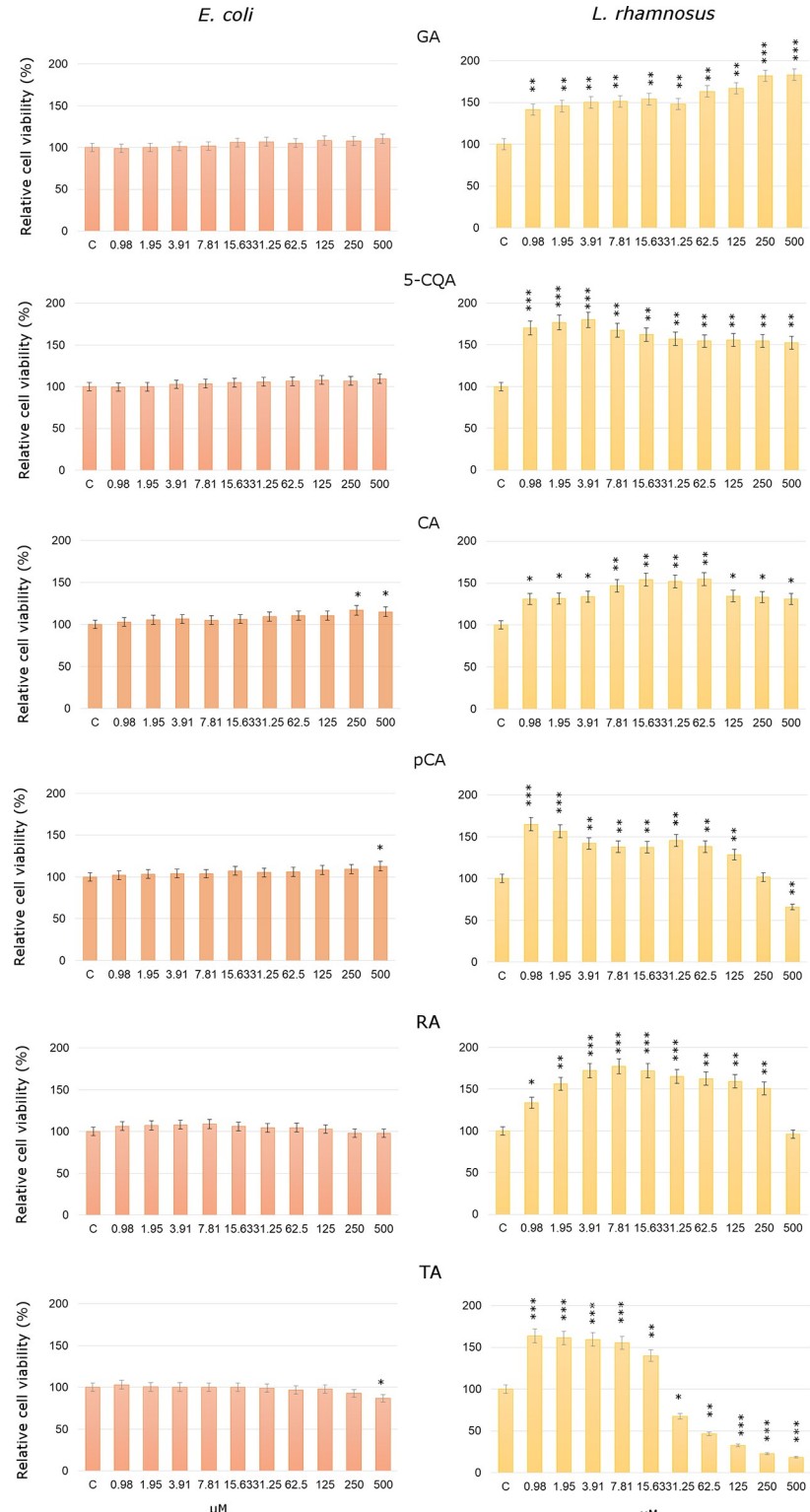

**Fig 3. Relative cell viability after GA, 5-CQA, CA, p-CA, RA and TA acids treatment after 24 h depending on the concentration of the tested compounds (\*- p<0.05, \*\*- p<0.01, \*\*\*- p<0.001 represent significant effects between treated and untreated bacteria followed by a Dunnett's test and error bars represent the standard deviation ±SD).**

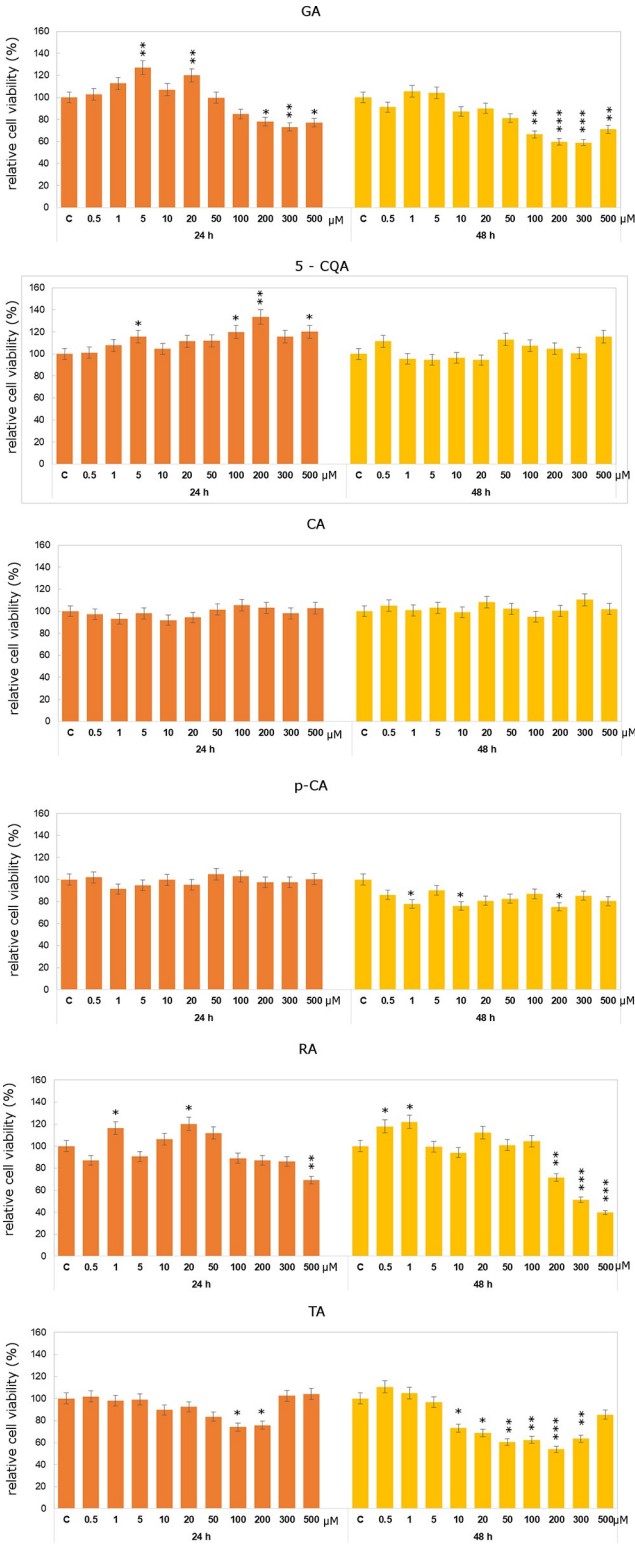

**Fig 4. Relative cell viability results for Caco-2 cells exposed to the range of 5-CQA, CA, p-CA, RA, GA and TA concentrations for 24h and 48h calculated as a percentage of control, untreated cells.** Each value on the graph is the mean of three independent experiments and error bars show the standard deviation (SD). * p < 0.05, ** p < 0.01 and *** p < 0.001 represent significant effects between treatments and control followed by a Dunnett's test.

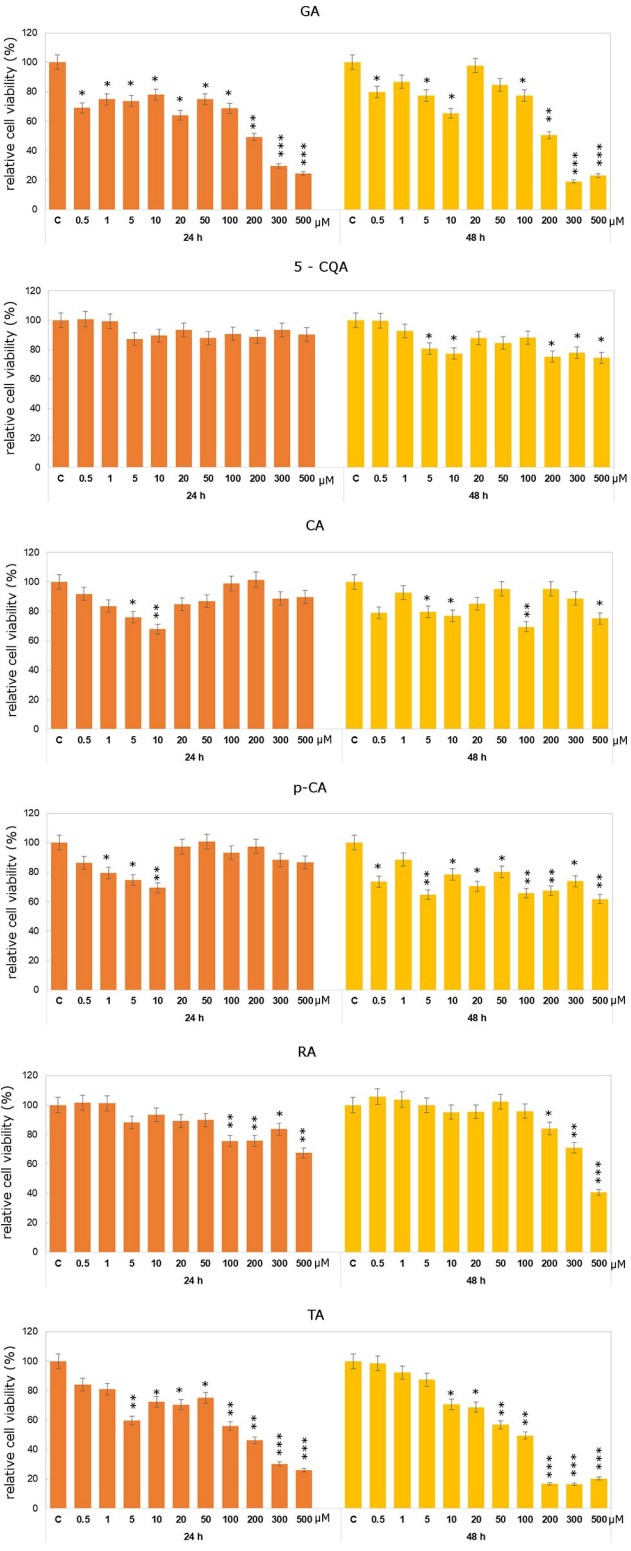

**Fig 5. Relative cell viability results for DLD-1 cells exposed to the range of 5-CQA, CA, p-CA and RA concentrations for 24h and 48h calculated as a percentage of control, untreated cells.** Each value on the graph is the mean of three independent experiments and error bars show the standard deviation (SD). * p < 0.05, ** p < 0.01 and *** p < 0.001 represent significant effects between treatments and control followed by a Dunnett's test.

presented in the literature, similarly to our results, indicate differences between the tested cell lines, which are partly due to differences in the genetic and metabolic profile of cancer cells [12, 18, 19]. The main difference between the Caco-2 and DLD-1 lines is their origin. Caco-2 cells represent Dukes' stage B, which means that the cancer has grown into the muscular layer of the intestine, while DLD-1 cells represent Dukes' stage C, which means that the cancer has metastasized to at least one lymph node in the intestinal area. By considering gene expression in both lines, differences between both lines were identified, which translate into their response to pro-apoptotic factors [66].

## Conclusions

In the study it was observed that the tested compounds did not affect the viability of *E.coli* in the tested concentration range (0.98–500 μM), which means that they do not stimulate the growth of a potentially pathogenic bacterium naturally present in human microbiome. Additionally, GA, CA, 5-CQA, p-CA, RA possessed prebiotic properties toward *L. rhamnosus* in the concentration range 0.98–500 μM. For TA the prebiotic effect was observed at the lower concentration range 0.98–15.63 μM. The lipophilicity (described by the logP parameter) of the phenolic compounds was correlated with their antibacterial activity toward *L. rhamnosus*. The compounds (i.e. TA, p-CA) with the highest values of logP exhibited an inhibitory effect on microbial growth at lower concentration levels than the phenolic compounds with the higher logP. What is interesting the calculated energy of HOMO orbital reflected changes in the lipophilicity of these compounds and may be used as a additional parameter to describe the biological activity of phenolic compounds toward *L. rhamnosus*. The cytotoxic activity of the phenolic acids was tested in the same concentration range as the antimicrobial. Based on the obtained results, it can be concluded that a significantly better response to the applied treatments was obtained in the case of the DLD-1 line than Caco-2, and the most effective in relation to both cell lines was TA. On the other hand, CA showed the lowest biological activity, especially in the Caco-2 line. The highest cytotoxic properties of TA toward DLD-1 cell line correspond with the highest lipophilicity and stability of the molecule compared to other tested phenolic compounds. Although these two parameters are important factors determining the biological activity of chemicals, the explanation of the high activity of TA is more complicated and is related to other mechanisms involved in induction of cell apoptosis like generation of reactive oxygen species, regulation of apoptotic and anti-apoptotic proteins, suppression and promotion of oncogenes or inhibition of matrix metalloproteinases. The aspect of possible induction of apoptosis is one of the most important elements of modern therapy from the point of view of eliminating cancer cells. The vast majority of the tested compounds seem to be promising potential anticancer agents, but they require further analysis at the molecular level to understand the mechanisms of their action.

## Supporting information

**S1 Fig. Co-occurrence of selected keywords in articles in 2014–2023 (search terms: "Phenolic compounds", "cytotoxicity", "antimicrobial activity"), created with VOSviewer [3].** (TIFF)

**S2 Fig.** Structures of *p*-coumaric acid (A), caffeic acid (B), rosmarinic acid (C), chlorogenic acid (D), gallic acid (E) and tannic acid (F). (TIF)

**S1 Table. Antimicrobial effects of phenolic acids on microbial strains (minimal inhibitory concentration (MIC) and inhibition zone (IZ) *MIC–minimal inhibitory concentration,**

IZ–inhibition zone, MBC–minimum bactericidal concentration, MQIC–minimal quorum inhibitory concentration, mm–milimeter, mM–milimolar concentration, %–percentage.
(PDF)

**S2 Table. Effects of phenolic acids on cell lines in in vitro assays.**
(PDF)

**S3 Table. Values of dipole moment and energy calculated for *p*-coumaric acid (*p*-CA), caffeic acid (CA), gallic acid (GA), chlorogenic acid (5-CQA), rosmarinic acid (RA) and tannic acid (TA).**
(DOCX)

## Author Contributions

**Conceptualization:** Monika Kalinowska, Włodzimierz Lewandowski.

**Data curation:** Monika Kalinowska, Maciej Kozłowski.

**Formal analysis:** Monika Kalinowska, Renata Świsłocka, Elżbieta Wołejko, Agata Jabłońska-Trypuć, Urszula Wydro, Maciej Kozłowski, Kamila Koronkiewicz.

**Funding acquisition:** Włodzimierz Lewandowski.

**Investigation:** Monika Kalinowska, Renata Świsłocka.

**Methodology:** Agata Jabłońska-Trypuć, Urszula Wydro.

**Visualization:** Agata Jabłońska-Trypuć, Kamila Koronkiewicz.

**Writing – original draft:** Monika Kalinowska, Elżbieta Wołejko, Agata Jabłońska-Trypuć, Urszula Wydro, Maciej Kozłowski, Kamila Koronkiewicz, Jolanta Piekut.

**Writing – review & editing:** Monika Kalinowska, Kamila Koronkiewicz, Włodzimierz Lewandowski.

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
