## [Decision Letter · Decision Letter 0]

20 Nov 2023

PONE-D-23-32977Structural characterization and evaluation of antimicrobial and cytotoxic activity of six plant phenolic acidsPLOS ONE

Dear Dr. Kalinowska,

Thank you for submitting your manuscript to PLOS ONE. After careful consideration, we feel that it has merit but does not fully meet PLOS ONE’s publication criteria as it currently stands. Therefore, we invite you to submit a revised version of the manuscript that addresses the points raised during the review process.

We look forward to receiving your revised manuscript.

Kind regards,

Jorddy Neves Cruz

Academic Editor

PLOS ONE

Journal Requirements:

- Biological Activity of New Cichoric Acid–Metal Complexes in Bacterial Strains, Yeast-Like Fungi, and Human Cell Cultures In Vitro - https://doi.org/10.3390/nu12010154

3. In your revision ensure you cite all your sources (including your own works), and quote or rephrase any duplicated text outside the methods section. Further consideration is dependent on these concerns being addressed.

National Science Centre research project number 2018/29/B/NZ9/01997

This research was founded by National Science Centre (NCN), Poland, under the research project number 2018/29/B/NZ9/01997.

National Science Centre research project number 2018/29/B/NZ9/01997

7. We note that you have stated that you will provide repository information for your data at acceptance. Should your manuscript be accepted for publication, we will hold it until you provide the relevant accession numbers or DOIs necessary to access your data. If you wish to make changes to your Data Availability statement, please describe these changes in your cover letter and we will update your Data Availability statement to reflect the information you provide.

Reviewers' comments:

Reviewer's Responses to Questions

**Comments to the Author**

1. Is the manuscript technically sound, and do the data support the conclusions?

Reviewer #1: Yes

Reviewer #2: Partly

Reviewer #3: Yes

Reviewer #4: Yes

2. Has the statistical analysis been performed appropriately and rigorously? 

Reviewer #1: Yes

Reviewer #2: I Don't Know

Reviewer #3: Yes

Reviewer #4: I Don't Know

3. Have the authors made all data underlying the findings in their manuscript fully available?

Reviewer #1: Yes

Reviewer #2: Yes

Reviewer #3: Yes

Reviewer #4: Yes

4. Is the manuscript presented in an intelligible fashion and written in standard English?

Reviewer #1: Yes

Reviewer #2: No

Reviewer #3: Yes

Reviewer #4: Yes

5. Review Comments to the Author

Reviewer #1: (1) Explanation for varying effects of compounds concentrations on cell viability, why lower concentrations of the some cases significantly increased cell viability, while higher concentrations decreased it?

(2) The data in Tables 4 and 5 have serious issues in various parts, such as incorrect values.

Reviewer #2: The work aims to do some computational calculation characterizing the chemical structures of selected phenolic acids and presents the results of antimicrobial and cytotoxic studies against two microbial strains and two cell lines.

General comments:

The manuscript is a mix of literature data and experimental results, however the submission is indicated as research work. In the research article the information provided on figure 1 is not suitable. The information provided in tables 4-5 is characteristic for review paper, not for research work. It is not clear what was a purpose of showing it.

The introduction section is far too long. It seems like a part from the review manuscript. Before the reader goes to the aim of the work is already overwhelmed by the number of mentioned studies. Also, since so many research was done on the selected phenolic acids, the novelty of this manuscript has to be much more explained and underlined. It is not clear what was a selection criteria to include these phenolic acids.

There is some potential in the selection of structures for the research. They differ by the number of factors, not only by the amount of hydroxyl groups. However this potential was not used for the benefit of this research. The chemical calculation performed are not correlated in the discussion with the observed biological activity. The ionization and lypophilicity influences the penetration of the compound into the cells, hence this information could help to explain the activity or lack of it, however this type of analysis of the generated data was not done. The results and discussion is rather a presentation of results and it lacks the attempt to explain the observations.

Other remarks:

LogP and pKa values of studied phenolic acids are available in the PubChem database, so what was a purpose to calculate them?

Table 4 and 5 are shown before tables 1-3.

What solvent was used to prepare working concentrations of studied phenolic acids in the antimicrobial test?

Table 6 does not show the units of the presented values. Some of the information presented in this table is already available in the literature, eg: https://doi.org/10.1002/bio.2932

Reviewer #3: The article is on the structural characterization and evaluation of the antimicrobial and cytotoxic activity of six plant phenolic acids. It is a well-designed article, fictionally. There are good results in the article. It has high citation potential.

This article can be publish with minor revision

But introduction is so long, please write short

Reviewer #4: The manuscript is well-written and presents a comprehensive analysis. However, I encountered difficulty in interpreting the figures (bar graph). The figures appear unclear, hindering a clear understanding of the presented data and its significance. It is crucial that the figures are enhanced for better visibility and comprehension.

6. PLOS authors have the option to publish the peer review history of their article (what does this mean?). If published, this will include your full peer review and any attached files.

Reviewer #1: No

Reviewer #2: No

Reviewer #3: **Yes: **Mustafa Sevindik

Reviewer #4: No

---

## [Author Response · Author response to Decision Letter 0]

12 Jan 2024

Please find enclosed the answers to reviewers’ comments. Thank you for the comments, suggestions that improved the manuscript.

Reviewer #1: (1) Explanation for varying effects of compounds concentrations on cell viability, why lower concentrations of the some cases significantly increased cell viability, while higher concentrations decreased it?

Answer: Thank you for the valuable comment. In both tested lines, decreases in cell viability were observed at higher concentrations of the analyzed compounds. However, in fact, a different response was observed for both tested lines at the lower concentrations analyzed. An increase in proliferation was observed in the Caco-2 line. This may be due to the fact that, although these are lines of colorectal adenocarcinoma, they are different from each other. The DLD-1 line is a cell line that is more tumorigenic and exhibited great chances of resistance and recurrence. However, the Caco-2 line is less tumorigenic. The different response to the applied compounds may result from differences in the structure and functioning of both cell lines. It may also be related to the chemical structure of the analyzed compounds. To explain why low concentrations of some of the tested compounds have a stimulating effect on the Caco-2 line, more thorough and detailed research should be carried out on the mechanisms of action of the tested compounds in both cell lines, which is in our scientific plans for the next experiment. The explanation was added to the manuscript.

(2) The data in Tables 4 and 5 have serious issues in various parts, such as incorrect values.

Answer: Table 4 and 5 have been corrected.

Reviewer #2: The work aims to do some computational calculation characterizing the chemical structures of selected phenolic acids and presents the results of antimicrobial and cytotoxic studies against two microbial strains and two cell lines.

General comments:

The manuscript is a mix of literature data and experimental results, however the submission is indicated as research work. In the research article the information provided on figure 1 is not suitable. The information provided in tables 4-5 is characteristic for review paper, not for research work. It is not clear what was a purpose of showing it.

Answer: Thank you for you valuable comments. Figure 1 as well as the tables 4 and 5 have been moved to the supplementary data. The Introduction has been significantly shortened.

The introduction section is far too long. It seems like a part from the review manuscript. Before the reader goes to the aim of the work is already overwhelmed by the number of mentioned studies. 

Answer: The Introduction has been significantly shortened.

Also, since so many research was done on the selected phenolic acids, the novelty of this manuscript has to be much more explained and underlined. It is not clear what was a selection criteria to include these phenolic acids.

Answer: Thank you for the comment, it helps us to improve the manuscript. The motivation for undertaking the research and the explanation of the selection of compounds are additionally included in the Introduction. “In this study, we will evaluate the antimicrobial activity of these six natural phenolic acids against a bacterial strains living in the digestive system, i.e. Escherichia coli and Lactobacillus rhamnosus based on 3-(4,5-dimethylthiazol-2-yl)-2,5-diphenyltetrazolium bromide colorimetric assay. Studies of antimicrobial activity aim to determine the suitability of the tested compounds as potential preservatives with antimicrobial properties against selected food pathogens (E.coli). Moreover, the studies on the effect of the analysed compounds on L. rhamnosus were conducted, which are part of the human intestinal microflora to exclude any negative effects on beneficial probiotic bacteria We will also investigate those phenolic acids cytotoxic effects of on two colorectal adenocarcinoma cell lines:Caco-2 and DLD-1 cell lines in order to analyze potential anticancer activity of studied compounds. Additionally, some structural parameters of the phenolic acids such as lipophilicity coefficient (logP) and reactivity of molecules on the basis of parameters calculated in Gaussian 09W program such aslike ionization potential, electron affinity, electronegativity, chemical hardness, chemical softness, dipole moment and electrophilicity index. The six natural phenolic compounds were chosen on purpose. They have proven biological properties such as antioxidant, antimicrobial or chemopreventive evaluated in different model systems of bacteria or cell lines.The chosen ligands are derivatives of two important aromatic acids, i.e. benzoic and cinnamic acids which differ in the presence of vinyl moiety between the aromatic ring and carboxylic group. Moreover they differ in the number of hydroxyl substituents in the ring (i.e. p-coumaric, caffeic and gallic acid), some of them are caffeic acid esters (i.e. rosmarinic and chlorogenic acids) or gallic acid ester (i.e. tannic acid). All this structural difference influence on the solubility, lipophilicity and reactivity of molecules. Therefore the following questions may arise: (1) does the number of -OH substituents in the aromatic ring is the main factor that affect the antimicrobial and cytotoxic activity of compounds (toward selected bacteria and cell lines), (2) does the conjugation of caffeic and gallic acid with other molecules influence on the physico-chemical and biological properties of molecules, (c) do the selected experimental (logP) and theoretical (e.g. energy of HOMO or LUMO orbitals) correlates with the biological activity of molecules?”

There is some potential in the selection of structures for the research. They differ by the number of factors, not only by the amount of hydroxyl groups. However this potential was not used for the benefit of this research. The chemical calculation performed are not correlated in the discussion with the observed biological activity. The ionization and lypophilicity influences the penetration of the compound into the cells, hence this information could help to explain the activity or lack of it, however this type of analysis of the generated data was not done. The results and discussion is rather a presentation of results and it lacks the attempt to explain the observations.

Answer: Thank you for the comment. The discussion was improved and the structure-activity discussion was in-depth, taking into account theoretical parameters and lipophilicity. The discussion is in the section Results and discussion, Antimicrobial activity, Cytotoxic activity and final remarks in Conclusions. 

Other remarks:

LogP and pKa values of studied phenolic acids are available in the PubChem database, so what was a purpose to calculate them? 

Answer: The table was removed to supplementary. Collecting data in one place shortens the time needed to search it.

Table 4 and 5 are shown before tables 1-3. 

Answer: It was corrected.

What solvent was used to prepare working concentrations of studied phenolic acids in the antimicrobial test?

Answer: The stock solution was prepared by dissolving it in TrisHCl buffer prior analysis.

Table 6 does not show the units of the presented values. Some of the information presented in this table is already available in the literature, eg: https://doi.org/10.1002/bio.2932

Answer: We agree. The table was removed to supplementary.

Reviewer #3: The article is on the structural characterization and evaluation of the antimicrobial and cytotoxic activity of six plant phenolic acids. It is a well-designed article, fictionally. There are good results in the article. It has high citation potential.

This article can be publish with minor revision. But introduction is so long, please write short

Answer: The Introduction has been significantly shortened.

Reviewer #4: The manuscript is well-written and presents a comprehensive analysis. However, I encountered difficulty in interpreting the figures (bar graph). The figures appear unclear, hindering a clear understanding of the presented data and its significance. It is crucial that the figures are enhanced for better visibility and comprehension.

Answer: Thank you for the comments. The figures were corrected.

Journal Requirements:

- Biological Activity of New Cichoric Acid–Metal Complexes in Bacterial Strains, Yeast-Like Fungi, and Human Cell Cultures In Vitro - https://doi.org/10.3390/nu12010154

Answer: Thank you for the comments. The experimental part, which is similar in both publications, has been corrected the to get rid of repetitions in the description of the methodology.

3. In your revision ensure you cite all your sources (including your own works), and quote or rephrase any duplicated text outside the methods section. Further consideration is dependent on these concerns being addressed.

Answer: Thank you. It was checked.

National Science Centre research project number 2018/29/B/NZ9/01997

Answer: It is correct, the funder (National Science Centre) had no role in study design, data collection and analysis, decision to publish, or preparation of the manuscript.

This research was funded by National Science Centre (NCN), Poland, under the research project number 2018/29/B/NZ9/01997.

National Science Centre research project number 2018/29/B/NZ9/01997

Answer: I understand. The information about funding was deleted from the Acknowledgement and removed to the Funding Statement. 

Answer: The data that support the findings of this study will be openly available in a repository. Unfortunately, we are currently still waiting to obtain a doi number. It will be added as soon as possible.

---

## [Editor Report · Decision Letter 1]

9 Feb 2024

Structural characterization and evaluation of antimicrobial and cytotoxic activity of six plant phenolic acids

PONE-D-23-32977R1

Dear Dr. Kalinowska,

We’re pleased to inform you that your manuscript has been judged scientifically suitable for publication and will be formally accepted for publication once it meets all outstanding technical requirements.

Kind regards,

Jorddy Neves Cruz

Academic Editor

PLOS ONE
---

## [Editor Report · Acceptance letter]

29 Apr 2024

PONE-D-23-32977R1 

PLOS ONE

Dear Dr. Kalinowska, 

I'm pleased to inform you that your manuscript has been deemed suitable for publication in PLOS ONE. Congratulations! Your manuscript is now being handed over to our production team.

Kind regards, 

on behalf of

Dr. Jorddy Neves Cruz 

Academic Editor

PLOS ONE